# Brief Communications: Observations of a Glacier Outburst Flood from Lhotse Glacier, Everest Area, Nepal

**David R. Rounce[1], Alton C. Byers[2], Elizabeth A. Byers[3], and Daene C. McKinney[1]**

[1] {Center for Research in Water Resources, University of Texas at Austin, Austin, TX, USA}

[2] {Institute of Arctic and Alpine Research, University of Colorado Boulder, Boulder, CO, USA}

[3]{Appalachian Ecology, Elkins, WV, USA}

Correspondence to: David R. Rounce (david.rounce@utexas.edu)

Keywords: Glacier outburst flood, Supraglacial lake, Englacial conduit, Everest, Nepal

## Abstract

Glacier outburst floods with origins from Lhotse Glacier, located in the Everest region of Nepal, occurred on 25 May 2015 and 12 June 2016. The most recent event was witnessed by investigators, which provided unique insights into the magnitude, source, and triggering mechanism of the flood. The field assessment and satellite imagery analysis following the event revealed that most of the flood water was stored englacial and the flood was likely triggered by dam failure. The flood's peak discharge was estimated to be 210 $m^3$ $s^{-1}$.

## 1 Introduction

Glacier outburst floods occur when stored glacier water is suddenly unleashed. Triggering mechanisms of these outburst floods include landslides, ice falls, and/or avalanches entering a proglacial lake resulting in a wave that overtops the dam leading to dam failure, dam failure due to settlement, piping, and/or the degradation of an ice-cored moraine, heavy rainfall that can alter the hydrostatic pressures placed on the dam, and many others (Richardson and Reynolds, 2000; Carrivick and Tweed, 2016). In the Himalaya, a specific subset of outburst floods called glacial lake outburst floods (GLOFs) has received the most attention with respect to hazards, likely because of their potentially large societal impact (e.g., Vuichard and Zimmermann, 1987). In contrast, glacier outburst floods in the Himalaya, herein referring to outburst floods that are not generated by a proglacial lake, have received relatively little attention likely due to their seemingly unpredictable nature, which has resulted in these events rarely being observed

(Fountain and Walder, 1998). While they are a known hazard and discussed in the literature (e.g., Richardson and Reynolds, 2000), few studies in Asia have investigated these hazards in detail (Richardson and Quincey, 2009).

Glacier outburst floods can occur sub-, en-, or supra-glacially when the hydrostatic pressure of the stored water exceeds the structural capacity of the damming body, when stored water is connected to an area of lower hydraulic potential, when englacial channels are progressively enlarged in an unstable manner, and/or when catastrophic glacier buoyancy occurs (Fountain and Walder, 1998; Richardson and Reynolds, 2000; Gulley and Benn, 2007). For debris-covered glaciers, the drainage of supraglacial ponds commonly occurs through englacial conduits, which facilitate connections to areas of lower hydraulic potential (Gulley and Benn, 2007). These englacial conduits develop on debris-covered glaciers in the Himalaya through cut-and-closure mechanisms associated with meltwater streams, the exploitation of high permeability areas that provide alternative pathways to the impermeable glacier ice, and through hydrofracturing processes (Gulley and Benn, 2007; Benn et al., 2009; Gulley et al., 2009a; Gulley et al., 2009b).

During the last half century, debris-covered glaciers in the Everest region have experienced significant mass loss (e.g., Bolch et al., 2011), which has led to the development of glacial lakes and supraglacial ponds (Benn et al., 2012). Proglacial lakes may develop if the surface gradient of the glacier is gentle (< 2°), while steeper gradients (> 2°) will help drain these ponds (Quincey et al., 2007). This causes supraglacial ponds to have large temporal and spatial variations as they frequently drain and fill (Horodyskyj, 2015; Miles et al., 2016; Watson et al., 2016). This drainage can occur on the glacier's surface and/or subsurface (Benn et al., 2012).

Lhotse Glacier (27°54'12" N, 86°52'40" E) is an avalanche-fed debris-covered glacier that extends 8.5 km from the peak of Lhotse at 8501 m to the glacier's terminus at 4800 m (Figure 1a). The lowest 3.5 km of the glacier is relatively stagnant and contains many supraglacial ponds. The upper 4 km, located beneath the headwall of Lhotse, is still quite active (Quincey et al., 2007), which can be seen by its highly crevassed features and its transient supraglacial ponds indicating frequent changes in the glacier's subsurface (Watson et al., 2016). Lhotse Glacier is one of the few glaciers in the region that lacks a steep bounding terminal moraine; instead, the terminus of the glacier is relatively steep (> 6°), which facilitates the drainage of supraglacial ponds and prevents the development of a large proglacial lake (Quincey et al., 2007). As these

supraglacial ponds drain and fill, they can cover up to 1.3-2.5% of the debris-covered glacier's
surface at any time (Watson et al., 2016). Speleological surveys conducted at Lhotse Glacier
found that cut-and-closure mechanisms and the exploitation of high permeability areas were the
main contributors to the development of englacial conduits and the drainage of supraglacial
ponds (Gulley and Benn, 2007).

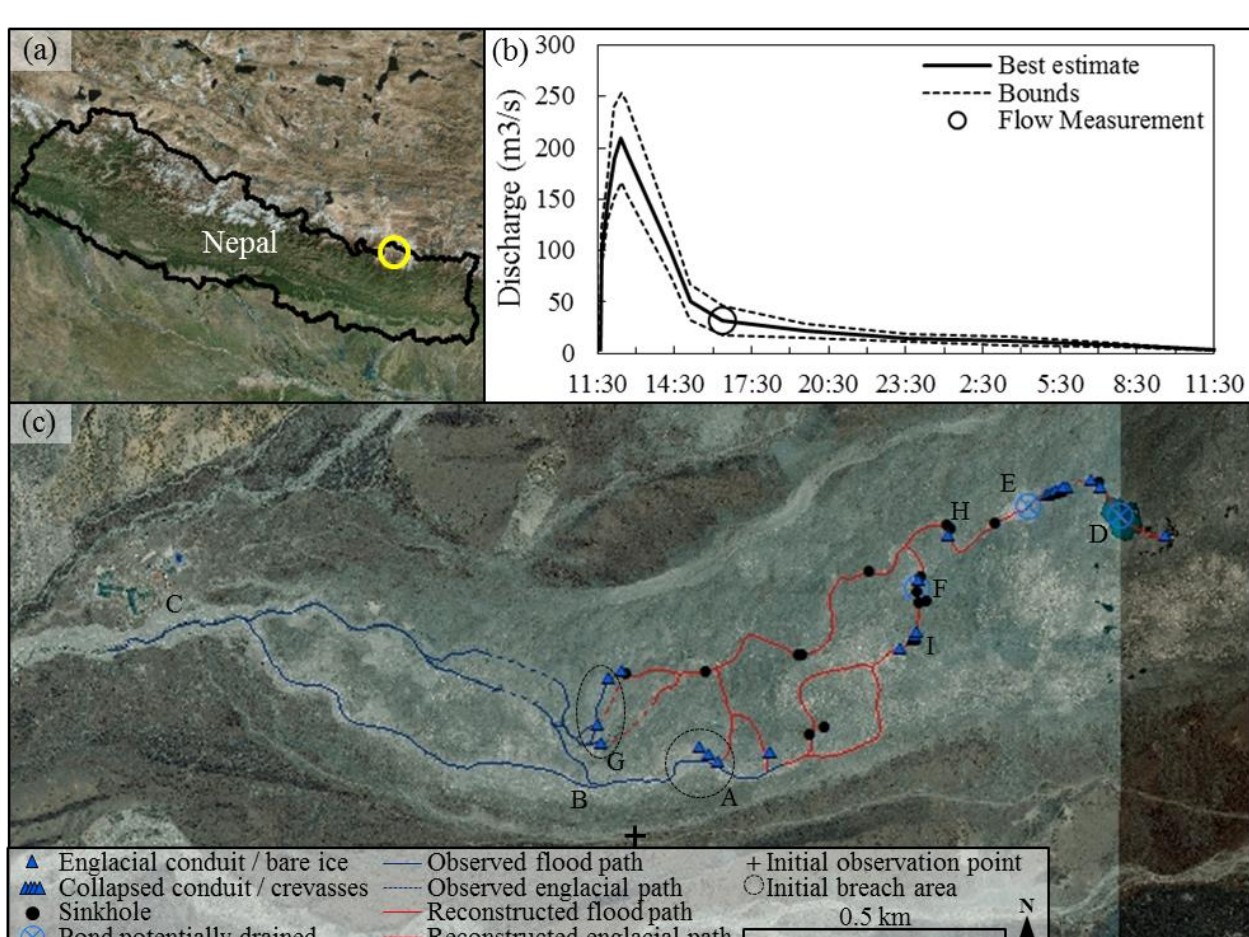

Figure 1. (a) Location of Lhotse Glacier in Nepal, (b) hydrograph of the glacier outburst flood
from Lhotse Glacier on 12 June 2016, and (c) map of observations and the reconstructed flood
path down to the village of Chukhung with letters corresponding to key features in Figure 2.
**2 Methods**
Glacier outburst floods with origins from Lhotse Glacier occurred on 12 June 2016 and 25 May
2015. The 2015 event was reported by local community members, while the 2016 event was
observed by the investigators from the southern lateral moraine of Lhotse Glacier (Figure 1c).
This provided a rare opportunity to photograph, record, and observe the outburst flood as it
unfolded. Flow measurements at 4:22 p.m., approximately four hours after the peak discharge,
were estimated from cross sectional areas and float velocities using bundles of sticks in a
relatively straight section of the channel below the village of Chukhung (27°54'03" N, 86°51'46"
E). Average velocity for the flow measurements was estimated to be 85% of the float velocity
(Rantz et al., 1982). Uncertainty associated with the flow measurements comprised errors in
river width (± 1 m), depths (± 0.3 m), float distance (± 1 m), and time (± 1 s). Peak flow was
conservatively estimated using the same average velocity with cross sectional areas derived from
high water marks.
During 14-21 June 2016, investigators conducted a field assessment on Lhotse Glacier to
reconstruct the flood path. Key features, which included bare ice faces, entrances and exits of
englacial conduits, sinkholes, collapsed tunnels, and ponds, were examined, photographed, and
measured using a handheld GPS (Garmin Montana) and a laser range finder (Nikon Forestry Pro).
Bio-indicators were also documented to assist reconstruction efforts. These indicators included
visual observations of recently uprooted and displaced alpine shrubs providing insight into the
surficial flood path. The presence of high water marks or wet, fine sediment that indicated
potential sinkholes or drained ponds were also recorded.
High resolution (0.5 m) satellite imagery (DigitalGlobe, Inc.) was used to assess the draining and
filling of supraglacial ponds around the 2015 and 2016 events based on manual delineations.
Specifically, imagery from 14 May 2016 (WorldView-2) and 29 October 2016 (WorldView-2)
were used to assess the 2016 event, and imagery from 08 May 2015 (GeoEye-1), 25 May 2015
(WorldView-2), and 07 June 2015 (WorldView-1) were used to assess the 2015 event. The
image from 14 May 2016 was also used as a background image for the reconstruction of the
2016 glacier outburst flood.
**3 Results**
**3.1 Direct observations:** At 11:40 a.m. on 12 June 2016, three landslide-like features began
flowing almost simultaneously down a south-facing slope of Lhotse Glacier, followed by large
amounts of discharging water from three apparent englacial conduits and one supraglacial stream
(Figure 1c, 2A). At the same time, approximately 200 m northwest of these landslide-like
features, large amounts of sediment-laden water was observed to be discharging into the main
channel from multiple englacial conduits and supraglacial channels (Figure 2B). Around 12:10

p.m., an additional supraglacial torrent and two supraglacial streams, located upglacier and to the east of the initial observations, joined the floodwater discharging from this initial area. The discharging water immediately began ponding and quickly breached the pond allowing the floodwater to propagate downstream and join the pre-existing main channel in addition to creating a secondary channel down the southern lateral moraine (Figures 1c, 2B). During this time, channel banks composed of ice and debris were severely undercut as the floodwater melted the surrounding ice as well.

The main channel continued to flow downstream until it re-entered englacial conduits (Figure 1c), which created an "ice bridge" that allowed investigators to cross the secondary and main channel after the peak flow started subsiding around 12:26 p.m. At 4:22 p.m., discharge below Chukhung was measured to be $32 \pm 14$ m$^3$ s$^{-1}$. Peak discharge was estimated retroactively to be $210 \pm 43$ m$^3$ s$^{-1}$. This estimate is considered to be conservative since it uses average velocity measurements taken four hours after peak discharge. Interestingly, this estimate agrees well with an empirical approach for predicting peak discharge based on glacier-bed area (Fountain and Walder, 1998), which predicts the peak discharge to be $38 - 1500$ m$^3$ s$^{-1}$ based on a glacier area of 6.825 km$^2$ for Lhotse Glacier (Arendt et al., 2015). A best-estimate hydrograph (Figure 1b) was reconstructed based on the photos of the water level at the ice bridge showing a peak flow of $210 \pm 43$ m$^3$ s$^{-1}$ at 12:26 p.m. followed by a gradual falling limb such that the discharge returned to normal conditions within 24 hours. The shape and timing of the hydrograph is consistent with the 1985 glacial lake outburst flood from Dig Tsho (Vuichard and Zimmerman, 1987), although the peak flow from Lhotse Glacier was significantly smaller. Based on this hydrograph, the overall flood volume was estimated to be $2.65 \times 10^6$ m$^3$ ($1.88 - 3.45 \times 10^6$ m$^3$ for the estimated low and high bounds, respectively). Minimal damage was caused to the community of Chukhung, which community members credited to the recently constructed gabions (Figure 2C). The main damage was the loss of a pedestrian bridge, an outbuilding, and small amounts of floodwater in the courtyard of one lodge. Supplementary material provides footage of the observed events.

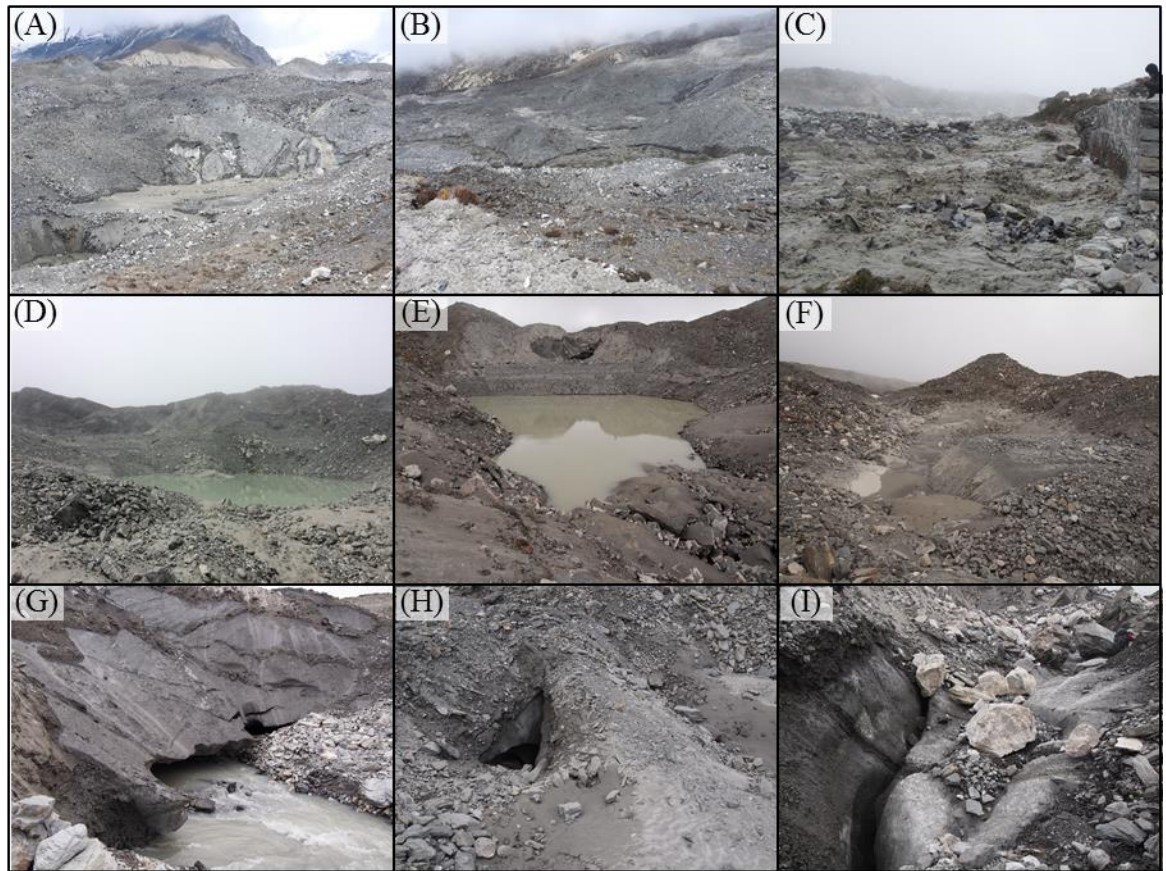

Figure 2. Key features of the glacier outburst flood from Lhotse Glacier: (A) subsurface and supraglacial flooding where the event was first observed, (B) main channels of flood path during the flood's peak, (C) flood undercutting the gabions at Chukhung, at 2:19 p.m., (D) potentially drained pond with large bare ice faces behind it, (E) potentially drained pond with a collapsed englacial conduit behind it, (F) potentially drained pond with sinkholes, (G) meltwater exiting the glacier into the main channel via a large englacial conduit, (H) a vertical englacial conduit and sinkholes with wet, fine sediment indicating a drainage pathway, and (I) large vertical crevasses with clean ice likely from the supraglacial flood path.

**3.2 Post-flood observations:** A detailed field assessment of Lhotse Glacier was conducted to reconstruct the glacier outburst flood by identifying potential flood pathways, englacial conduits, sinkholes, and drained ponds (Figure 1c). Satellite imagery from 14 May 2016 revealed a sizeable supraglacial pond (27°54'20" N, 86°53'27" E) with an area of 4900 m$^2$ located directly beneath a large bare ice face (~10-20 m) that was considerably smaller during our field assessment (Figure 2D). This pond also had fine, wet sediment along its slopes in addition to a series of bare ice, sinkholes, and englacial conduits located immediately downstream, which could have facilitated its drainage. This was the pond located the furthest upglacier that

appeared to have recently drained, although a detailed assessment of all the supraglacial ponds
and terrain upglacier was not possible due to time limitations.
This ponded water likely entered a series of englacial conduits and potentially supraglacial
pathways before entering another supraglacial pond located ~200 m down-glacier (Figure 1c).
This second supraglacial pond had similar indicators of having recently drained (Figure 2E),
although the satellite image does not show a large supraglacial pond.  It is possible that
meltwater filled the pond between the time that the satellite image was acquired and the glacier
outburst flood.  A collapsed englacial conduit was observed between these two ponds (Figure 1c)
in addition to a series of sinkholes along with an entrance to an englacial conduit located
immediately downstream of the pond (Figure 2H).  Based on recently uprooted and displaced
alpine shrubs, the flood appeared to continue downstream where it branched into multiple paths
(Figure 1c).  The southern branch appears to have entered a third supraglacial pond (Figure 2F),
which had similar indicators and large sinkholes.  Downstream of this third pond was a small
valley that was littered with areas of clean ice and deep crevasses (Figure 2I).  It appears that this
supraglacial pathway and englacial conduits fed into the flood torrent that joined the initial
discharge at 12:10 p.m. (Figure 1c).  The other branch showed signs of supraglacial and englacial
pathways in the form of bio-indicators, sinkholes, and englacial conduits as well, which appear
to have contributed to the heavy flow that was observed discharging into the main channel as
well (Figure 2G).
**3.3 Satellite imagery analysis:** Satellite imagery provides unique opportunities to observe the
contribution of supraglacial ponds to these glacier outburst flood events; however, it is important
that this imagery is acquired immediately before and after the event as these supraglacial ponds
experience large temporal and spatial changes (Figure 3).  In order to estimate the potential flood
volume associated with the drainage of supraglacial ponds, an area-to-volume relationship was
used (Cook and Quincey, 2015).  Based on the change in areal extent between 14 May 2016 and
29 October 2016, the drained volume from the furthest supraglacial pond upglacier (Figure 1c,
Figure 2D) was $0.01 \times 10^{6}$ m$^{3}$.  This volume is two orders of magnitude less than the estimated
flood volume of $2.65 \times 10^{6}$ m$^{3}$, which suggests that the drainage of a single supraglacial pond
contributes very little to the overall flood volume.  In fact, if all of the 274 supraglacial ponds
(0.21 km$^{2}$) that were present on Lhotse Glacier on 14 May 2016 drained completely, the
potential flood volume would only be 0.52 x $10^6$ m$^3$. This provides strong evidence that a
significant amount of the flood water was stored in the glacier's subsurface.

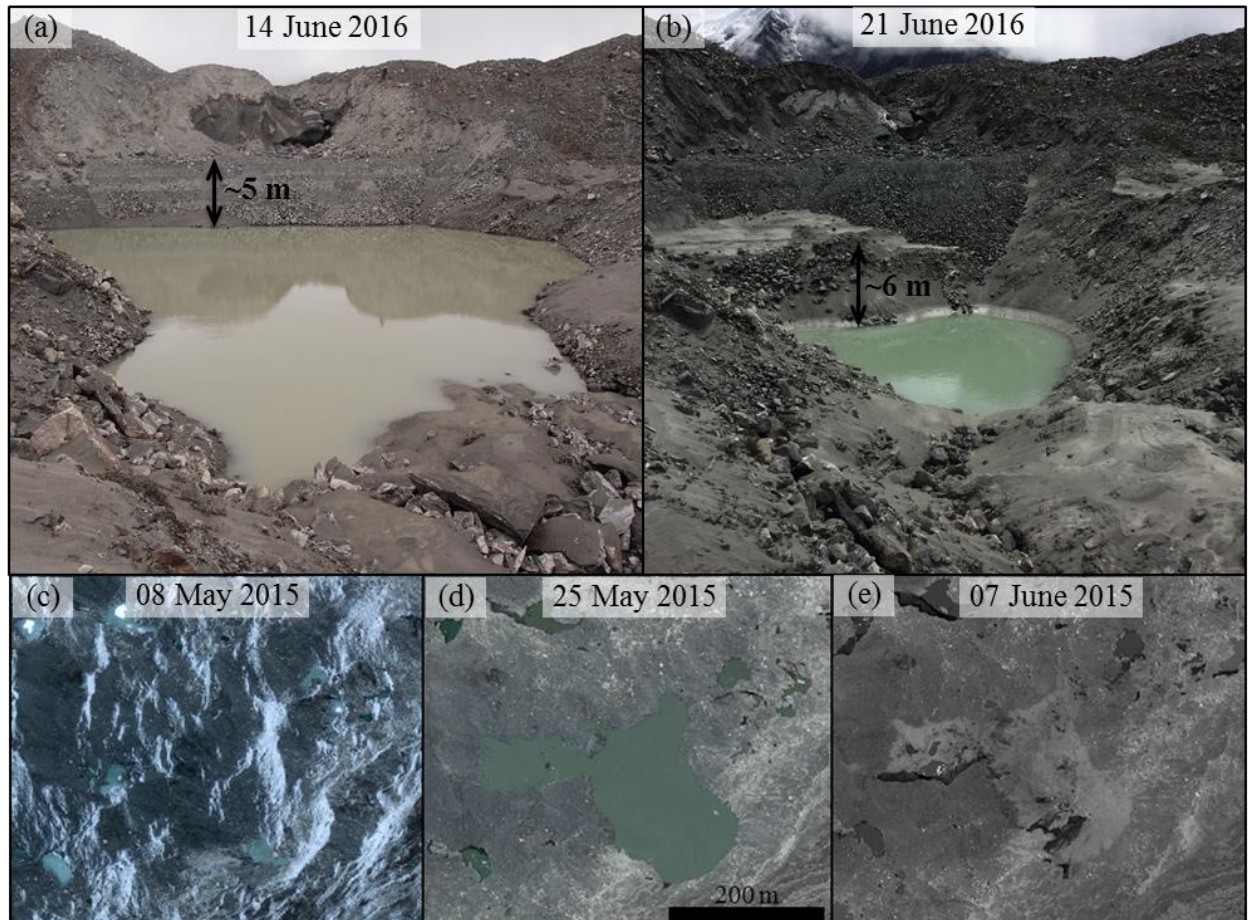

Figure 3. Images showing the temporal changes of supraglacial ponds (a, b) following the 2016
glacier outburst flood and (c, d, e) around the 2015 glacier outburst flood.
The glacier outburst flood on 25 May 2015 also originated from Lhotse Glacier and occurred
overnight (Sherpa, L., personal communication, 09 June 2015). Satellite imagery from 08 May
2015, 25 May 2015, and 07 June 2015 reveals a large supraglacial pond (0.036 km$^2$) filling
between 08 – 25 May and draining completely between 25 May – 07 June (Figure 3c, d, e). The
drainage of this supraglacial pond could have contributed up to 0.17 x $10^6$ m$^3$ to the 2015 glacier
outburst flood. Community members reported that the 2016 event was larger than the 2015
event. A similar outburst event was also reported to have occurred in early May 2016 in the
vicinity of the "crampon put-on point" (5600 m) of Island Peak (6189 m) that damaged sections
of the high and low basecamp regions (Sherpa, P.T., personal communication, 18 June 2016).

## 4 Discussion

**4.1 Source of the flood water:** The field observations immediately following the 2016 glacier outburst flood suggest that some of the source water was from the drainage of supraglacial ponds; however, the satellite imagery analysis revealed that the drainage of supraglacial ponds alone could not account for the entire flood volume. Therefore, the water that was unleashed during the 2016 glacier outburst flood was likely stored in both the glacier's subsurface and in supraglacial ponds. Once the flood was initiated, the melting of ice from both the channel banks and in the englacial conduits caused these outlet pathways to grow, which likely contributed more water to the total flood volume in addition to opening more efficient pathways for the stored water to drain.

**4.2 Triggering mechanisms:** Potential triggering mechanisms for these glacier outburst floods include dam failure, the rapid drainage of stored lake water through hydraulically efficient pathways, and/or catastrophic glacier buoyancy. The sudden discharge observed during the 2016 event (Figure 1b) suggests that the trigger was most likely dam failure or the rapid drainage of stored lake water, since catastrophic glacier buoyancy typically has a hydrograph with a more gradual rising limb (Fountain and Walder, 1998).

Dam failure would require an englacial conduit to be temporarily blocked, which could occur if meltwater refroze in the conduits over the winter (Gulley et al., 2009) or if passage closure processes caused an englacial conduit to close (Benn et al., 2012). The former blockage scenario seems more likely since these glacier outburst floods have occurred in back-to-back years and the refreezing of meltwater is an annual process. During the early melt season the subsurface drainage system is distributed and inefficient, which provides opportunities for water to accumulate englacial (Fountain and Walder, 1998). Dam failure may then occur if the hydrostatic pressures in the englacial conduits exceed the cryostatic pressure that was previously constraining the stored water thereby causing the dam to rupture (Richardson and Reynolds, 2000). Alternatively, as water accumulates in the englacial conduits, the changes in water pressure can cause these conduits to grow in an unstable manner thereby causing drainage to occur (Fountain and Walder, 1998). This progressive enlargement is similar to piping failures and the failures of ice dammed lakes (Richardson and Reynolds, 2000).

The rapid drainage of stored lake water through hydraulically efficient pathways is another plausible triggering mechanism that commonly occurs for supraglacial ponds in the Everest region (Benn et al., 2012). Field observations of supraglacial ponds (Figure 2D, E) revealed that there were englacial conduits located at the end of both of these lakes that likely helped facilitate their drainage. This link between the englacial conduits and supraglacial ponds is not surprising as near-surface water storage on glaciers can result from water accumulating in englacial conduits (Fountain and Walder, 1998). Once these ponds come in contact with an englacial conduit or a highly permeable layer, the warm pond water can cause significant internal ablation that helps facilitate the drainage of additional stored water. The drainage of supraglacial ponds that was observed for the 2015 and 2016 events supports this theory; however, as previously discussed, the drainage of supraglacial ponds alone likely accounts for a small fraction of the total flood volume.

This suggests that the most feasible triggering mechanism is likely some form of dam failure resulting from the material blocking the englacial conduits being overburdened or failure resulting from the progressive enlargement of englacial conduits. The timing of these events, which occurred around the start of the monsoon season, further supports this triggering mechanism as this provides ample time for these englacial conduits to fill with meltwater or precipitation prior to dam failure. It should not come as a surprise that this time of year is also when supraglacial pond cover is at its highest (Miles et al., 2016) as this may be indicative of the amount of water stored englacial as well. In fact, it is possible that the large supraglacial pond that filled immediately before the 2015 glacier outburst flood (Figure 3c, d) was the surficial expression of the englacial conduits accumulating too much water, which could explain the pond's short lifespan once the englacial conduits drained. This may also explain how the second supraglacial pond (Figure 1c, 2E) was not apparent in satellite imagery on 24 May 2016, but appeared to have drained recently based on field observations (Figure 3a, b), i.e., the pond likely filled between 24 May 2016 and the glacier outburst flood. On the other hand, the most upglacier supraglacial pond (Figure 1c, 2D) was present in the imagery and had been growing since 2011 (Watson et al., 2016), which indicates that the rapid drainage of supraglacial ponds through hydraulically efficient pathways may also be contributing to these glacier outburst floods as well, albeit contributing a smaller volume than the water stored englacial.

## 5 Conclusions

The direct observations of the glacier outburst flood on 12 June 2016 from Lhotse Glacier provide unique insight into the magnitude, source, and trigger mechanisms associated with these rarely observed events. The flood occurred suddenly and reached a peak discharge of 210 $\text{m}^3 \text{ s}^{-1}$ only 45 minutes after the flood began. The detailed field assessment conducted in the days immediately following the event in conjunction with the satellite imagery analysis was used to determine that most of the flood water originated from the glacier's subsurface. Based on the sudden discharge and magnitude of the event, the flood appeared to be triggered by dam failure due to the englacial conduits rupturing from being overburdened or from the englacial conduits progressively enlarging in an unstable manner until failure occurred. Community members reported that another glacier outburst flood originating from Lhotse Glacier occurred on 25 May 2015, which suggests that Lhotse Glacier may provide unique opportunities to study these complex events in more detail in the future. Future work should seek to improve our understanding of the triggering mechanisms and size of these events through detailed field surveys assessing both the glacier's surface and subsurface combined with methodically tasked high resolution satellite imagery. This work is necessary as improving our understanding of the frequency and magnitude of these events has important economic and social implications for downstream communities and hydropower companies.

**Acknowledgements**

The authors acknowledge the support of the NSF-CNH program (award no. 1516912) for the support of David Rounce, Alton Byers, and Daene McKinney. Dhananjay Regmi of Himalayan Research Expeditions provided important logistical support during fieldwork. Bidhya Sharma provided additional images and videos for this study. The authors would also like to thank Duncan Quincey and the anonymous reviewer for their comments that helped improve the manuscript.

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

**Supplementary Material**

Video footage of the glacier outburst flood from 12 June 2016 may be found at http://www.crwr.utexas.edu/video/Lhotse_Flood_Supplement_V3.mp4.