# Peer review of "Brief Communications: Observations of a Glacier Outburst Flood"

_The Cryosphere, 2016_

## Referee Comment (RC1) · Anonymous Referee #1 · 6 Dec 2016

This Brief Communication describes outburst flood initiation by drainage through englacial conduits, a process that has been inferred from observations on debris covered glaciers in the Everest region, but rarely observed. As such, this communication makes an important contribution to the growing literature on outburst floods and I think publication as a communication is appropriate. I have a few general comments that I think should be addressed in a revision.

I am aware of few observations of englacial outburst floods, which is the primary reason why I think this Brief Communication should be published. Please highlight this important facet of the flood in both the title and abstract. One possible suggestion for a title would be: Observations of the role of englacial conduits in a Glacier Outburst

[Figure]

Flood from the Lhotse Glacier, Everest Area, Nepal.

Abstracts should present the key findings of the research rather than tell the reader to read the article. I suggest starting over from scratch.

Page 1, Line 7: Does the paper need to mention that the results of this paper are not the opinion of the WV DEP? Page 1, Line 18: the location of "unleashing" is not downstream. Page 1, Line 19: change "mass movement" to landslides, ice falls and/or avalanches. Also, this will sound picky, but the cause of the flood is the resulting wave that overtops the dam, leading to failure. The other triggers should probably also be described in the context of how they contribute to dam failure. Page 2: Line 25: Not clear why supraglacial ponds are indicative of active ice dynamics. Page 3, Line 10: The rationale for assuming that average velocity was 85% of the float velocity is?

Page 7, line 13: hydrostatic pressure exceeding cryostatic pressure seems an unlikely trigger for an englacial/supraglacial lake drainage mechanism. More plausible is that two lake basins at different elevations became connected by a permeable feature within the ice (such as a relic supraglacial channel; Benn et al., 2012 or Gulley and Benn, 2007). Line 15: It is not clear what is meant by "open up outlets of lower hydraulic potential" Page 8, lines 16-19: I don't think that two events in two years can be called repetitive. Page 8, line 21: I think the authors need to clarify that this is possibly the first time that an englacial outburst flood has been witnessed. I'm not aware of any of similar observations on debris covered glaciers. Page 8, line 26: The authors have not presented any direct evidence that the subglacial drainage system played a role in this flood. Figure 1: Is there no way to create a DEM of the glacier surface? It would go a long way towards showing supraglacial flow paths.

---

## Referee Comment (RC2) · D.J. Quincey (Referee) · 6 Dec 2016

This manuscript presents the results of opportunistic observations and measurements conducted around the Lhotse Glacier during and after a glacier outburst flood in the pre-monsoon season of 2016. The aim is to provide some insight into the magnitude of the event and the processes that triggered the flood as well as controlled the progression of flood waters through the supraglacial and englacial hydrological systems. Field observations are supported by some basic satellite image analysis, and the main conclusion of the work is that there is still much to be learned about these rarely observed events. This holds true even for this manuscript; as the reader I was not entirely sure what I had learned at the end of it other than the basic information relating to the

flood. Nevertheless, given that such events are very rarely observed in real-time, I am strongly in favour of the observations being published. I do have several suggestions that may strengthen the manuscript further, and some less major comments that the authors should address.

Major comments

The main deficiency of the submission is that no substantial conclusion relating to the source or trigger mechanism of the flood can be reached based on the data that are presented. Three additional analyses may provide some further illumination:

1. Can you explore the satellite image archives (even just Google Earth) to see whether the large supraglacial pond (at D in your Figure 1) has persisted over several years, or whether it was a new feature in the build-up to the flood? If it was new, it lends support to your interpretation that the flood was related to recent meltwater storage, possibly from a blockage in the englacial system, with the pond representing the surface expression at the head of the stored water. If it is not new, then this interpretation is less likely to be valid. Its disappearance would point more towards more 'normal' drainage as the pond intercepts an englacial channel. If you integrate these observations with the next suggestion, you might at least be able to say with more certainty whether the water was supraglacially or englacially sourced.

2. Can you use your rough estimates of discharge to back-calculate a conservative overall flood volume? Clearly there will be large uncertainty associated with the calculation, but it may be sufficient to rule out the simple drainage of one or two supraglacial ponds if, as I suspect is the case, the overall flood discharge exceeds what you might reasonably expect the combined supraglacial pond volume to be. Then you might be able to say with some certainty whether water was being stored beneath the glacier surface.

3. Are there any reports from locals about the shape of the hydrograph? Or are your own observations sufficient to say anything about that? On page 7, line 11, the

text states that the flood had sudden onset. If so, this implies that there was a sudden failure of the dam rather than something more gradual like surface water tapping into an inefficient hydrological system. If even a crude hydrograph shape can be reconstructed it may help you at least narrow down the flood trigger.

I suspect that one or more of these extra analyses will allow you to rule out a few hypotheses, even if they don't get you to a conclusive interpretation.

Minor comments

P1 Abstract: this needs some work. The abstract should summarise what was done and what was found out.

P1 26-27: isn't the lack of attention simply because these floods are so unpredictable and thus rarely observed?

P2 3-6: these are a mixture of cause (triggers) and effect of drainage. Channels becoming progressively enlarged, for example, are not a cause of floods. They are possibly a control on the discharge, and are certainly more of an effect of the flood than a cause.

P2 15: insert reference regarding mass loss

P2 20: are subsurface and englacial not the same thing?

P2 27: worth mentioning here that Lhotse is one of the few glaciers in the region without a steep bounding terminal moraine (i.e. that might trap or at least modulate flood waters in other locations)?

P3 9 (and elsewhere) is it Chukung, or Chukhung? I've seen both, but I think the latter is more common?

P3 11: replace 'accounts for' with 'comprises'?

P3 14: missing 'the'

P3 22: missing 'the'

P4 20: 'retrospectively?

P5 20: can you give an idea of the pond size? Just its rough diameter as measured from the satellite data would be helpful

P6 6-7: change to 'was not possible due to...'?

P7 3-4: I'm not convinced your observations reveal anything about the triggers in the current version of the manuscript so you might choose to rephrase this sentence

Figure 3: can you indicate the scale that is shown here?

P7 11-13: what does this sentence actually mean? That hydrofracture was the cause? Or that a dam was breached? And what is the evidence? If you are suggesting that the englacial hydrology was blocked then you need to state this more clearly.

P 7 11: was the outburst definitely sudden? If so, you have evidence of dam failure and you may be able to infer something more about the trigger than you already have.

P7 18: do conduits 'rupture' in this way? I'm not familiar with this if so...

P7 19-20: wouldn't a simpler explanation be that the englacial system was overwhelmed so the water found another (i.e. surface) route?

P7 10-22: It might be helpful to separate out the discussion of the triggers vs subsequent processes as they are very different.

P8 1-2: the increase in discharge is more likely to be related to the ability of the developing channels to convey water, don't you think?

P8 3: what do you mean by meltwater storage in this context? Englacial specifically? Can you clarify?

P8 13-16 this process is not normally sudden. I think you have to invoke a slightly different chain of processes.

P8 22: I don't think this is likely to be true. Partly it depends on what you class as a scientist (locals can also be 'scientists') and many 'scientists' have been working in the Himalaya for many years. I suggest removing this statement as it is not necessary and it is unsubstantiated.

P8 23: as the manuscript is presented I don't think you shed any light on the triggers, so you might want to modify this.

P8 26-27: do you mean the difficulty of making interpretations on limited data highlights the lack of knowledge? Can you clarify?

---

## Author Comment (AC1) · 17 Jan 2017

**Authors' Response to Reviewers' Comments:**
**Brief Communications: Observations of a Glacier Outburst Flood from Lhotse**
**Glacier, Everest Area, Nepal" by Rounce et al.**

We would like to thank both Duncan Quincey and the other anonymous reviewer for their insightful and constructive comments. The following response seeks to address all of their comments and detail the subsequent revisions made to the text.

**Response to Duncan Quincey's Comments**

Major Comments

The main deficiency of the submission is that no substantial conclusion relating to the source or trigger mechanism of the flood can be reached based on the data that are presented. Three additional analyses may provide some further illumination:

> A detailed response to each of the three analyses is provided below. These additional analyses added significant detail to the conclusions regarding the source and triggering mechanism of the flood such that a new section was added to the results concerning satellite imagery and the discussion was completely rewritten. Similarly, the abstract and conclusions were rewritten to reflect these changes as well. Specific details for each of these sections is provided in the comments below.

(1) Can you explore the satellite imagery archives (even just GoogleEarth) to see whether the large supraglacial pond (at D in your Figure 1) has persisted over several years, or whether it was a new feature in the build-up to the flood? If it was new, it lends support to your interpretation that the flood was related to recent meltwater storage, possibly from a blockage in the englacial system, with the pond representing the surface expression at the head of the stored water. If it is not new, then this interpretation is less likely to be valid. Its disappearance would point more towards more 'normal' drainage as the pond intercepts an englacial channel. If you integrate these observations with the next suggestion, you might at least be able to say with more certainty whether the water was supraglacially or englacially sourced.

> Both GoogleEarth and DigitalGlobe imagery reveal that the large supraglacial pond (D in Figure 1; herein referred to as supraglacial pond D) and the smaller supraglacial pond (E in Figure 1; herein referred to as supraglacial pond E) appear to drain and fill over time. WorldView-2 imagery from 14 May 2016 and 29 October 2016 show the areal extent of supraglacial pond was greatly reduced from 4900 $m^2$ to 1500 $m^2$, respectively. Based on Watson et al. (2016) who assessed the frequency of ponds using high resolution satellite imagery from 2002, 2011, 2013, and 2015, supraglacial pond D appears to have originated in 2011 where it was 360 $m^2$ and proceeded to grow to 1500 $m^2$ in 2013 and to 6500 $m^2$ in 2015. This analysis concerning the ponds surrounding the 2016 event has been added to the results section:

**"3.3 Satellite imagery analysis:** Satellite imagery provides unique opportunities to observe the contribution of supraglacial ponds to these glacier outburst flood events; however, it is important that this imagery is acquired immediately before and after the event as these supraglacial ponds experience large temporal and spatial changes (Figure 3). In order to estimate the potential flood volume associated with the drainage of supraglacial ponds, an area-to-volume relationship was used (Cook and Quincey, 2015). Based on the change in areal extent between 14 May 2016 and 29 October 2016, the drained volume from the furthest supraglacial pond upglacier (Figure 1c, Figure 2D) was $0.01 \times 10^6$ m$^3$. This volume is two orders of magnitude less than the estimated flood volume of $2.65 \times 10^6$ m$^3$, which suggests that the drainage of a single supraglacial pond contributes very little to the overall flood volume. In fact, if all of the 274 supraglacial ponds (0.21 km$^2$) that were present on Lhotse Glacier on 14 May 2016 drained completely, the potential flood volume would only be $0.52 \times 10^6$ m$^3$. This provides strong evidence that a significant amount of the flood water was stored in the glacier's subsurface."

Satellite imagery surrounding the 2015 GLOF event also reveals supraglacial ponds filling and draining. Specifically, a comparison of a WorldView-2 image from 25 May 2015 and a WorldView-1 image from 07 June 2015 shows a 36000 m$^2$ supraglacial pond (27.910°N, 86.907°E) that completely drained over this time period. Interestingly, a GeoEye-1 image from 08 May 2015 shows that this pond did not exist at that time and the frequency analysis done by Watson et al. (2016) also do not identify any large supraglacial ponds in this vicinity. This analysis concerning the ponds surrounding the 2015 event has also been added to the satellite imagery analysis results section:

"The glacier outburst flood on 25 May 2015 also originated from Lhotse Glacier and occurred overnight (Sherpa, L., personal communication, 09 June 2015). Satellite imagery from 08 May 2015, 25 May 2015, and 07 June 2015 reveals a large supraglacial pond (0.036 km$^2$) filling between 08 – 25 May and draining completely between 25 May – 07 June (Figure 3c, d, e). The drainage of this supraglacial pond could have contributed up to $0.17 \times 10^6$ m$^3$ to the 2015 glacier outburst flood. Community members reported that the 2016 event was larger than the 2015 event. A similar outburst event was also reported to have occurred in early May 2016 in the vicinity of the "crampon put-on point" (5600 m) of Island Peak (6189 m) that damaged sections of the high and low basecamp regions (Sherpa, P.T., personal communication, 18 June 2016)."

Unfortunately, the high resolution imagery that was available for the 25 May 2015 glacier outburst flood is not available for the 12 June 2016. The best imagery surrounding the 2016 event is from 14 May 2016 and 29 October 2016 as previously mentioned. Watson et al. (2016) and Miles et al. (2016) showed that supraglacial ponds frequently drain and fill over the course of a melt season, which makes it difficult to confidently determine which ponds may have drained during the 2016 event based on the available high resolution satellite imagery. Fortunately, the field observations described in the discussion paper can supplement the satellite images.

These field observations indicated that supraglacial ponds D & E likely drained on or around the 12 June 2016 glacier outburst flood as described in the initial manuscript (P5, L19 – P6, L15).

The similarities between the size and the timing of the 2015 and 2016 events based on reports from local residents indicate that these two glacier outburst floods were triggered by similar mechanisms. In fact, the 2015 event showed the complete drainage of a supraglacial pond that was an order of magnitude larger than those observed in 2016 (36000 $m^2$ vs. 3400 $m^2$, respectively), yet the 2016 event was larger according to local residents. This supports that hypothesis that the flood was related to the accumulation of meltwater storage despite the fact that supraglacial pond D was not necessarily a "new" pond. The reviewer suggests two explanations: (1) the flood was related to a blockage in the englacial system or (2) the flood is related to the normal drainage as the pond intercepts an englacial channel. We find it is difficult to differentiate these two processes from one another as there is little information on what normal discharge is or what constitutes a new pond and instead suggest that this flood is likely a combination of both processes, i.e., the drainage network during the early melt season may be distributed and inefficient, which causes the meltwater to accumulate until the glacier outburst flood releases the water and opens up new efficient channels as previously described in the paper. We have added the following explanation to the discussion regarding triggering mechanisms discussing this:

> "The rapid drainage of stored lake water through hydraulically efficient pathways is another plausible triggering mechanism that commonly occurs for supraglacial ponds in the Everest region (Benn et al., 2012). Field observations of supraglacial ponds (Figure 2D, E) revealed that there were englacial conduits located at the end of both of these lakes that likely helped facilitate their drainage. This link between the englacial conduits and supraglacial ponds is not surprising as near-surface water storage on glaciers can result from water accumulating in englacial conduits (Fountain and Walder, 1998). Once these ponds come in contact with an englacial conduit or a highly permeable layer, the warm pond water can cause significant internal ablation that helps facilitate the drainage of additional stored water. The drainage of supraglacial ponds that was observed for the 2015 and 2016 events supports this theory; however, as previously discussed, the drainage of supraglacial ponds alone likely accounts for a small fraction of the total flood volume.

> This suggests that the most feasible triggering mechanism is likely some form of dam failure resulting from the material blocking the englacial conduits being overburdened or failure resulting from the progressive enlargement of englacial conduits. The timing of these events, which occurred around the start of the monsoon season, further supports this triggering mechanism as this provides ample time for these englacial conduits to fill with meltwater or precipitation prior to dam failure. It should not come as a surprise that this time of year is also when supraglacial pond cover is at its highest (Miles et al., 2016) as this may be indicative of the amount of water stored englacial as well. In fact, it is possible that

the large supraglacial pond that filled immediately before the 2015 glacier outburst flood (Figure 3c, d) was the surficial expression of the englacial conduits accumulating too much water, which could explain the pond's short lifespan once the englacial conduits drained.  This may also explain how supraglacial pond E (Figure 1c) was not apparent in satellite imagery on 24 May 2016, but appeared to have drained recently based on field observations (Figure 3a, b), i.e., the pond likely filled between 24 May 2016 and the glacier outburst flood.  On the other hand, supraglacial pond D (Figure 1c) was present in the imagery and had been growing since 2011 (Watson et al., 2016), which indicates that the rapid drainage of supraglacial ponds through hydraulically efficient pathways may also be contributing to these glacier outburst floods as well, albeit contributing a smaller volume than the water stored englacial."

(2) Can you use your rough estimates of discharge to back-calculate a conservative overall flood volume?  Clearly there will be large uncertainty associated with the calculation, but it may be sufficient to rule out the simple drainage of one or two supraglacial ponds if, as I suspect is the case, the overall flood discharge exceeds what you might reasonably expect the combined supraglacial pond volume to be.  Then you might be able to say with some certainty whether water was being stored beneath the glacier surface.

Based on photos of the water level before it re-entered the englacial conduits as discussed in the text (P4, L17-19) a best-estimate hydrograph was re-constructed (Figure R1).  The peak discharge from these photographs occurred at 12:26 p.m. and was estimated to be 210 $m^3$ $s^{-1}$.  Figure R1 shows the flow steeply increased during the first 30 minutes of the flood event and lasted for approximately 5 hours.  The shape and timing of this hydrograph is consistent with the constructed hydrograph for the Dig Tsho glacial lake outburst flood in 1985 (Vuichard and Zimmerman, 1987), although the peak flow from Lhotse Glacier was significantly smaller.  Based on this hydrograph, the overall flood discharge was estimated to be 2.65 x $10^6$ $m^3$ (1.88 – 3.45 x $10^6$ $m^3$ for the low and high estimates, respectively).

[Figure]

Figure R1. Estimate of the flood hydrograph from Lhotse Glacier 12 June 2016.

Fountain and Walder (1998) present an empirical equation for the magnitude of the peak flow from glacier outburst floods based on "glaciological experience" as follows:

$$Q_{MAX} = \frac{2Ad}{\tau}$$

where A is the glacier-bed area, d is the equivalent water layer over the enter glacier bed (~10 – 100 mm), and $\tau$ is the period of time over which the stored water is typically released (~15-60 min). The area of Lhotse Glacier according to GLIMS V5 is 6.825 km$^2$, which would estimate the magnitude of peak flow from Lhotse Glacier to range from 38 m$^3$ s$^{-1}$ to 1500 m$^3$ s$^{-1}$. While this is an empirical equation, it does lend confidence our estimate of peak flow. Furthermore, the timing of the peak flow, which occurred ~45 minutes after the flood was initiated, agrees with the timing of the typical release.

In order to estimate the potential flood volume associated with the drainage of supraglacial ponds, an area-to-volume relationship was used based on Cook and Quincey (2015). Based on the change in areal extent between 14 May 2016 and 29 October 2016, the drained volume associated with supraglacial pond D was 0.0107 x 10$^6$ m$^3$. This volume is two orders of magnitude less than the estimated flood volume, which supports the hypothesis that accumulated water in Lhotse Glacier's subsurface was an important source of flood water. In fact, in the 14 May 2016 image 274 supraglacial ponds were identified that covered an area of 0.21 km$^2$. If all of these ponds completely drained, which is very conservative, the total drained volume would only be 0.52 x 10$^6$ m$^3$. This total drained volume is still significantly smaller than the flood volume estimate of the glacier outburst flood (2.65 x 10$^6$ m$^3$), which provides strong evidence that subglacial discharge had a critical role in these glacier outburst flood events.

Figure R1 has been added to Figure 1 and part of the direct observations section of the results has been revised to the following to include the results of the hydrograph:

"The main channel continued to flow downstream until it re-entered englacial conduits (Figure 1c), which created an "ice bridge" that allowed investigators to cross the secondary and main channel after the peak flow started subsiding around 12:26 p.m. At 4:22 p.m., discharge below Chukhung was measured to be 32 ± 14 m$^3$ s$^{-1}$. Peak discharge was estimated retroactively to be 210 ± 43 m$^3$ s$^{-1}$. This estimate is considered to be conservative since it uses average velocity measurements taken four hours after peak discharge. Interestingly, this estimate agrees well with an empirical approach for predicting peak discharge based on glacier-bed area (Fountain and Walder, 1998), which predicts the peak discharge to be 38 – 1500 m$^3$ s$^{-1}$. A best-estimate hydrograph (Figure 1b) was reconstructed based on the photos of the water level at the ice bridge showing a peak flow of 210

$\pm$ 43 m$^3$ s$^{-1}$ at 12:26 p.m. followed by a gradual falling limb such that the discharge returned to normal conditions within 24 hours. The shape and timing of the hydrograph is consistent with the 1985 glacial lake outburst flood from Dig Tsho (Vuichard and Zimmerman, 1987), although the peak flow from Lhotse Glacier was significantly smaller. Based on this hydrograph, the overall flood volume was estimated to be 2.65 x 10$^6$ m$^3$ (1.88 – 3.45 x 10$^6$ m$^3$ for the estimated low and high bounds, respectively). Minimal damage was caused to the community of Chukhung, which community members credited to the recently constructed gabions (Figure 2C). The main damage was the loss of a pedestrian bridge, an outbuilding, and small amounts of floodwater in the courtyard of one lodge. Supplementary material provides footage of the observed events."

(3) Are there any reports from locals about the shape of the hydrograph? Or are your own observations sufficient to say anything about that? On page 7, line, 11, the text states that the flood had sudden onset. If so, this implies there was a sudden failure of the dam rather than something more gradual like surface water tapping into an inefficient hydrological system. If even a crude hydrograph shape can be reconstructed it may help you at least narrow down the flood trigger.

> The shape of the hydrograph was estimated in response to the second comment based on photos near the ice bridge and direct observations by the authors (P4, L17-19). The shape clearly depicts that the flood was a sudden event, which, as the reviewer suggests, indicates there was a sudden failure within the glacier's subsurface as opposed to more gradual processes of the surface water slowly finding more efficient channels. A discussion concerning the timing and size of the flood and the information that it provides concerning the triggering mechanisms has been added to the text in place of the previous discussion:

> > **"4.2 Triggering mechanisms:** Potential triggering mechanisms for these glacier outburst floods include dam failure, the rapid drainage of stored lake water through hydraulically efficient pathways, and/or catastrophic glacier buoyancy. The sudden discharge observed during the 2016 event (Figure 1b) suggests that the trigger was most likely dam failure or the rapid drainage of stored lake water, since catastrophic glacier buoyancy typically has a hydrograph with a more gradual rising limb (Fountain and Walder, 1998).

> > Dam failure would require an englacial conduit to be temporarily blocked, which could occur if meltwater refroze in the conduits over the winter (Gulley et al., 2009) or if passage closure processes caused an englacial conduit to close (Benn et al., 2012). The former blockage scenario seems more likely since these glacier outburst floods have occurred in back-to-back years and the refreezing of meltwater is an annual process. During the early melt season the subsurface drainage system is distributed and inefficient, which provides opportunities for water to accumulate englacial (Fountain and Walder, 1998). Dam failure may then occur if the hydrostatic pressures in the englacial conduits exceed the cryostatic pressure that was previously constraining the stored water thereby causing the dam to rupture

(Richardson and Reynolds, 2000). Alternatively, as water accumulates in the englacial conduits, the changes in water pressure can cause these conduits to grow in an unstable manner thereby causing drainage to occur (Fountain and Walder, 1998). This progressive enlargement is similar to piping failures and the failures of ice dammed lakes (Richardson and Reynolds, 2000)."

Minor Comments

P1 Abstract: this needs some work. The abstract should summarize what was done and what was found out.

The abstract has been completely redone:

"Glacier outburst floods with origins from Lhotse Glacier, located in the Everest region of Nepal, occurred on 25 May 2015 and 12 June 2016. The most recent event was witnessed by investigators, which provided unique insights into the magnitude, source, and triggering mechanism of the flood. The field assessment and satellite imagery analysis following the event revealed that most of the flood water was stored englacial and the flood was likely triggered by dam failure. The flood's peak discharge was estimated to be 210 $m^3$ $s^{-1}$."

P1 26-27: isn't the lack of attention simply because these floods are so unpredictable and thus rarely observed?

Yes, this is likely true as well. The sentence has been changed to the following:

"In contrast, glacier outburst floods in the Himalaya, herein referring to outburst floods that are not generated by a proglacial lake, have received relatively little attention likely due to their seemingly unpredictable nature, which has resulted in these events rarely being observed (Fountain and Walder, 1998)."

P2 3-6: these are a mixture of cause (triggers) and effect of drainage. Channels becoming progressively enlarged, for example, are not a cause of floods. They are possibly a control on the discharge, and are certainly more of an effect of the flood than a cause.

The authors agree that the channels becoming progressively enlarged that was observed during the 2016 glacier outburst flood was an effect of the flood as opposed to a cause. However, Richardson and Reynolds (2000) state that "there are three recorded mechanisms by which glacier outburst floods occur: the rupture of an internal water pocket, the progressive enlargement of internal drainage channels and catastrophic glacier buoyancy, or 'jacking', with sub-glacial discharge". Similarly, Fountain and Walder (1998) discuss the enlargement of internal drainage channels as a means of rapidly draining stored glacier water. As this section of the paper is meant to introduce the existing knowledge regarding these glacier outburst floods, we have elected to keep the progressive enlargement of an englacial conduit as a triggering

mechanism; however, we have slightly altered the sentence to clarify the enlargement of englacial conduits versus the drainage channel enlargement that we observed during the event. The sentence now reads:

"Glacier outburst floods can occur sub-, en-, or supra-glacially when the hydrostatic pressure of the stored water exceeds the structural capacity of the damming body, when stored water is connected to an area of lower hydraulic potential, when englacial channels are progressively enlarged in an unstable manner, and/or when catastrophic glacier buoyancy occurs (Fountain and Walder, 1998; Richardson and Reynolds, 2000; Gulley and Benn, 2007)."

We would also like to note that the observed channel enlargement from the 2016 event (P4, L8-10) is clearly stated as a cause of the flood and not as a triggering mechanism.

P2 15: insert reference regarding mass loss

Reference to Bolch et al. (2011) has been inserted as an example. Benn et al. (2012) also summarizes mass loss studies in the Everest region.

P2 20: are subsurface and englacial not the same thing?

Yes, this was a typo. The sentence was meant to refer to surficial, englacial, and subglacial drainage. The text has been revised to read "the glacier's surface and/or subsurface".

P2 27: worth mentioning here that Lhotse is one of the few glaciers in the region without a steep bounding terminal moraine (i.e. that might trap or at least modulate flood waters in other locations).

The authors agree this would be good information to include. The sentence has been revised to read as follows: "Lhotse Glacier is one of the few glaciers in the region that lacks a steep bounding terminal moraine; instead, the terminus of the glacier is relatively steep (> 6°)…"

P3 9 (and elsewhere) is it Chukung, or Chukhung? I've seen both, but I think the latter is more common?

The spelling has been changed in all cases to "Chukhung".

P3 11: replace 'accounts for' with 'comprises'?

This change has been made.

P3 14: missing 'the'

'the' has been inserted.

P3 22: missing 'the'

'the' has been inserted.

P4 20: 'retrospectively'?

This has been changed to 'retroactively'.

P5 20: can you give an idea of the pond size? Just its rough diameter measured from the satellite data would be helpful.

The area is 4900 m$^2$ based on satellite imagery from 14 May 2016. Its area has been added in the satellite imagery analysis results.

P6 6-7: change to 'was not possible due to…'?

This change has been made.

P7 3-4: I'm not convinced your observations reveal anything about the triggers in the current version of the manuscript so you might choose to rephrase this sentence.

Based on the response to previous comments and changes made to the manuscript, this study is able to discuss the triggers in more detail, which has been added to the paper as detailed in the response to the major comments.

Figure 3: can you indicate the scale that is shown here?

Yes, the image has been revised to include an approximate scale.

P7 11-13: what does this sentence actually mean? That hydrofracture was the cause? Or that a dam was breached? And what is the evidence? If you are suggesting that englacial hydrology was blocked then you need to state this more clearly.

The sentence was meant to discuss that dam failure likely occurred in the englacial conduits when the hydrostatic pressure exceeded the cryostatic pressure that was holding the water back, i.e., that a dam was breached. The discussion of the triggering mechanisms has been completely rewritten to reflect the response to the major comments above and we believe the various scenarios are very clearly discussed.

P7 11: was the outburst definitely sudden? If so, you have evidence of dam failure and you may be able to infer something more about the trigger than you already have.

Yes, this outburst sudden (Figure R1), which supports that there was some form of dam failure as a triggering mechanism as the reviewer suggest. This is reflected in the new discussion as previously discussed.

P7 18: do conduits 'rupture' in this way? I'm not familiar with this if so…

Based on the additional analysis into triggering mechanisms, we agree with the reviewer that this situation likely did not occur and therefore have removed it from the text.

P7 19-20: wouldn't a simpler explanation be that the englacial system was overwhelmed so the water found another (i.e. surface) route?

Yes, we agree with the reviewer's suggested explanation that the surface flow is caused when the englacial system was overwhelmed thereby giving the water an alternative pathway; however, this has been removed in the revised discussion.

P7 10-22: It might be helpful to separate out the discussion of the triggers vs subsequent processes as they are very different.

The discussion has been separated into two subsections: source of the flood water and triggering mechanisms.

P8 1-2: the increase in discharge is more likely to be related to the ability of the developing channels to convey water, don't you think?

Yes, we agree with the reviewer that the englargement of the channels and englacial conduits would also help convey water more efficiently. The sentence has been changed to the following:

"Once the flood was initiated, the melting of ice from both the channel banks and in the englacial conduits caused these outlet pathways to grow, which likely contributed more water to the total flood volume in addition to opening more efficient pathways for the stored water to drain."

P8 3: what do you mean by meltwater storage in this context? Englacial specifically? Can you clarify this?

This sentence was referring to flood water that was stored in the glacier's subsurface. Based on the additional analyses conducted in response to previous comments, there is clear evidence that this is important and the language used to describe the stored water has been clarified.

P8 13-16: this process is not normally sudden. I think you have to invoke a slightly different chain of processes.

The reviewer is correct that the evolution of the subglacial hydrological system in the Arctic does not occur suddenly, but changes over the course of the melt season. The citation is meant to show the similarity between an evolving subglacial hydrological system that transforms from an inefficient to an efficient system over time. This citation has actually been removed from the text and a better citation (Fountain and Walder, 1998) has been added, which deals with alpine glaciers. Furthermore, the text has been clarified to simply state that "During the early melt season the subsurface drainage system is distributed and inefficient, which provides opportunities for water to accumulate englacial."

P8 22: I don't think this is likely to be true. Partly it depends on what you class as a scientist (locals can also be 'scientists') and many 'scientists' have been working in the Himalaya for many years. I suggest removing this statement as it is not necessary and it is unsubstantiated.

This sentence has been removed.

P8 23: as the manuscript is presented I don't think you shed any light on the triggers, so you might want to modify this.

Based on the additional analyses included in response to both reviewers comments, we believe that the manuscript now sheds greater light on the source and potential triggers. The sentence now reads "… which provides unique insight into the magnitude, source, and triggering mechanisms of these events."

P8 26-27: do you mean the difficulty of making interpretations on limited data highlights the lack of knowledge? Can you clarify?

This sentence has been removed in the revised conclusions.

**Response to Anonymous Referee #1's Comments**

I am aware of few observations of englacial outburst flood, which is the primary reason why I think this Brief Communication should be published. Please highlight this important facet of the flood in both the title and abstract. One possible suggestion for a title would be: Observations of the role of englacial conduits in a Glacier Outburst Flood from the Lhotse Glacier, Everest Area, Nepal.

We appreciate the reviewers support and agree that the observations that we saw regarding the englacial nature of this flood are important; however, both satellite and field observations show that water stored in supraglacial ponds was likely a source of flood water as well. Similarly, the reconstruction of the flood path shows both supraglacial and englacial paths. Therefore, we believe that altering the title to focus solely on the role of englacial conduits is too focused and prefer not to change the title. The import role that these englacial had in the flood is detailed in both the abstract and the rest of the paper.

Abstracts should present the key findings of the research rather than tell the reader to read the article. I suggest starting over from scratch.

The abstract has been re-written as follows:

"Glacier outburst floods with origins from Lhotse Glacier, located in the Everest region of Nepal, occurred on 25 May 2015 and 12 June 2016. The most recent event was witnessed by investigators, which provided unique insights into the magnitude, source, and triggering mechanism of the flood. The field assessment and satellite imagery analysis following the event revealed that most of the flood water was stored englacial and the flood was likely triggered by dam failure. The flood's peak discharge was estimated to be 210 $m^3$ $s^{-1}$."

Page 1, Line 7: Does the paper need to mention that the results of this paper are not the opinion of the WV DEP?

The affiliation of Elizabeth Byers has been changed to Applachian Ecology, which does not require a similar statement.

Page 1, Line 18: the location of "unleashing" is not downstream.

The word 'downstream' has been removed.

Page 1, Line 19: change "mass movement" to landslide, ice falls and/or avalanches.

This change has been made.

Also, this will sound picky, but the cause of the flood is the resulting wave that overtops the dam, leading to failure. The other triggers should probably also be described in the context of how they contribute to dam failure.

The sentence has been revised to include the details of each triggering process as follows:

"Triggering mechanisms of these outburst floods include landslides, ice falls, and/or avalanches entering a proglacial lake resulting in a wave that overtops the dam leading to dam failure, dam failure due to settlement, piping, and/or the degradation of an ice-cored moraine, and heavy rainfall that can alter the hydrostatic pressures placed on the moraine, and many others (Richardson and Reynolds, 2000; Carrivick and Tweed, 2016)."

Page 2, Line 25: Not clear why supraglacial ponds are indicative of active ice dynamics.

The draining and filling of supraglacial ponds indicate that the subsurface of the glacier is changing. The sentence has been changed to "The upper 4 km, located

beneath the headwall of Lhotse, is still quite active (Quincey et al., 2007), which can be seen by its highly crevassed features and its transient supraglacial ponds indicating frequent changes in the glacier's subsurface (Watson et al., 2016).

Page 3, Line 10: The rationale for assuming that the average velocity is 85% of the float velocity is?

For natural channels the mean velocity is commonly estimated as 85% of the surface velocity based on the assumption of a logarithmic profile (Rantz et al., 1982). This source has been added.

Page 7, Line 13: Hydrostatic pressure exceeding cryostatic pressure seems an unlikely trigger for an englacial/supraglacial lake drainage mechanism. More plausible is that two lake basins at different elevations became connected by a permeable feature within the ice (such as a relec supraglacial channel; Benn et al., 2012 or Gulley and Benn, 2007).

We agree that the connection of two lake basins at different elevations via a permeable feature within the ice is a plausible and common scenario for the drainage of supraglacial ponds; however, the analysis that was done in response to the other reviewers comment indicates that the drainage volume from supraglacial ponds accounts for a small fraction of the total flood volume. Furthermore, the sudden nature of the flood (Figure R1 above) suggests that there was some form of dam failure. The satellite imagery suggests that both dam failure and the connection of lake basins at different elevations may have occurred. The discussion regarding triggering mechanisms has been edited to reflect this. Please see the response to the other reviewer's major comments for more detail.

Page 7, Line 15: It is not clear what is meant by "open up outlets of lower hydraulic potential"

This was meant to refer to the connection of two lake basins as the reviewer discusses in the previous comment. The language in the discussion has been changed to "the rapid drainage of stored lake water through hydraulically efficient pathways."

Page 8, Lines 16-19: I don't think that two events in two years can be called repetitive.

The use of repetitive has been removed from the text.

Page 8, Line 21: I think the authors need to clarify that this is possibly the first time that an englacial outburst flood has been witnessed. I'm not aware of any similar observations on debris-covered glaciers.

This sentence has been deleted entirely in response to a comment from the other reviewer.

Page 8, Line 26: The authors have not presented any direct evidence that the subglacial drainage system played a role in this flood.

> The satellite imagery analysis that was included in response to the other reviewer's comments clearly shows that the subglacial drainage system plays a role in this flood.

Figure 1: Is there no way to create a DEM of the glacier surface? It would go a long way towards showing supraglacial flow paths.

> Unfortunately, a high resolution DEM ($< 5$ m) of the glacier surface is not currently available for Lhotse Glacier around the time of the flood event. However, the authors did walk the flood path with a handheld GPS system, which helped determine the reconstruction efforts along with the bio-indicators. The recommendation to have high resolution imagery over this region (which could include repeat DEMs) has been included in a sentence concerning future work in the conclusions as follows:

> > "Future work should seek to improve our understanding of the triggering mechanisms and size of these events through detailed field surveys assessing both the glacier's surface and subsurface combined with methodically tasked high resolution satellite imagery."

**References** (new to the discussion paper)

Bolch, T., Pieczonka, T., and Benn, D.I.: Multi-decadal mass loss of glaciers in the Everest area (Nepal Himalaya) derived from stereo imagery, The Cryosphere, 5, 349-358, doi:10.5194/tc-5-349-2011, 2011.

Cook, S.J. and Quincey, D.J.: Estimating the volume of Alpine glacial lakes, Earth Surface Dynamics, 3:559-575, doi:10.5194/esurf-3-559-2015, 2015.

Miles, E.S., Willis, I.C., Arnold, N.S., Steiner, J., and Pellicciotti, F.: Spatial, seasonal and interannual variability of supraglacial ponds in the Langtang Valley of Nepal, 1999-2013, J. Glaciol., 1-18, doi:10.1017/jog.2016.120, 2016.

Rantz, S.E.: Measurement and computation of streamflow: volume 1, measurement of stage and discharge, No. 2175, USGPO, 1982.